# Global antibiotic dosing strategies in hospitalised children: Characterising variation and implications for harmonisation of international guidelines

**Michelle N. Clements**[1]*, **Neal Russell**[2], **Julia A. Bielicki**[2], **Sally Ellis**[3], **Silke Gastine**[4], **Yingfen Hsia**[2¤], **Joseph F. Standing**[4], **A. Sarah Walker**[1], **Mike Sharland**[2]

**1** MRC Clinical Trials Unit at UCL, London, United Kingdom, **2** Paediatric Infectious Diseases Research Group, Institute for Infection and Immunity, St George's University, London, United Kingdom, **3** Global Antibiotic Research & Development Partnership, Geneva, Switzerland, **4** Infection, Immunity and Inflammation, Institute of Child Health, University College London, London, United Kingdom

¤ Current address: School of Pharmacy, Queen's University, Belfast, United Kingdom
* michelle.clements@ucl.ac.uk

**Data Availability Statement:** All relevant data are within the paper and its Supporting Information files.

## Abstract

### Background

Paediatric global antibiotic guidelines are inconsistent, most likely due to the limited pharmacokinetic and efficacy data in this population. We investigated factors underlying variation in antibiotic dosing using data from five global point prevalence surveys.

### Methods & findings

Data from 3,367 doses of the 16 most frequent intravenous antibiotics administered to children 1 month–12 years across 23 countries were analysed. For each antibiotic, we identified standard doses given as either weight-based doses (in mg/kg/day) or fixed daily doses (in mg/day), and investigated the pattern of dosing using each strategy. Factors underlying observed variation in weight-based doses were investigated using linear mixed effects models. Weight-based dosing (in mg/kg/day) clustered around a small number of peaks, and all antibiotics had 1–3 standard weight-based doses used in 5%-48% of doses. Dosing strategy was more often weight-based than fixed daily dosing for all antibiotics apart from teicoplanin, which had approximately equal proportions of dosing attributable to each strategy. No strong consistent patterns emerged to explain the historical variation in actual weight-based doses used apart from higher dosing seen in central nervous system infections, and lower in skin and soft tissue infections compared to lower respiratory tract infections. Higher dosing was noted in the Americas compared to the European region.

### Conclusions

Antibiotic dosing in children clusters around a small number of doses, although variation remains. There is a clear opportunity for the clinical, scientific and public health communities

**Funding:** Funding/Support: This study was funded by GARDP. GARPEC was funded by the PENTA Foundation. MC and ASW are supported by core support from the Medical Research Council UK to the MRC Clinical Trials Unit [MC_UU_12023/22]. ASW is an NIHR Senior Investigator. The views expressed are those of the author(s) and not necessarily those of the NHS, the NIHR, or the Department of Health. This analysis was facilitated by Global Antibiotic R&D Partnership with funding from German Federal Ministry of Health, Médecins Sans Frontières, Netherlands Ministry of Health, Welfare and Sport, and UK Department for International Development Role of Funder/Sponsor (if any): SE contributed to the design of data collection tools for the GARPEC project, and is a co-author.

**Competing interests:** The authors have declared that no competing interests exist.

**Abbreviations:** FDD, fixed daily dose; PKPD, pharmacokinetic-pharmacodynamic; PPS, point prevalence survey; TDM, therapeutic drug monitoring; WBD, weight-based dose.

to consolidate behind a consistent set of global antibiotic dosing guidelines to harmonise current practice and prioritise future research.

## Introduction

Paediatric antibiotic dosing guidelines in children are inconsistent, at both international and national levels [1, 2], most likely due to the lack of pharmacokinetic/pharmacodynamic (PKPD) data in this population [1]. Large differences in the choice of antibiotic prescribed for the same indication have been observed in both children and neonates [3–5] between countries [6, 7], between hospitals within a country [8] and between doctors within a single hospital [9]. However, despite differences in PK, leading to different optimal doses in children compared to adults [10], and critically ill children compared to well children [11], little attention to date has been focused on quantifying the differences and identifying the factors underlying the variation observed in dosing.

Optimal dosing of antibiotics in children differs due to PKPD factors such as age, weight, or comorbidities of the individual, the therapeutic index or PKPD target parameter of the antibiotic, and species, site of infection, or resistance profile (both phenotyped and anticipated from local knowledge) of the organism [10]. Antibiotic dosing decisions may be based on either fixed daily doses (FDD), commonly split into age or weight bands, often in mg/day, or actual weight-based doses (WBD), often in mg/kg/day [12]. Studies in the UK and France have found substantial variation in vancomycin and gentamicin dosing practice in neonates [13, 14], a European study identified marked variation in WBD in children in two antibiotics [12], and a study in Pakistan found that less than 40% of children were receiving rational dosing for antibiotics prescribed in hospital [15]. Very few studies have attempted to identify the factors underlying clinicians' decisions to use specific doses and strategies. Here we used data from five global point prevalence surveys to investigate variation in dosing for 16 antibiotics given intravenously in hospital for treatment in children. We investigate the frequency of administration, examine whether dosing strategy is guided by FDD or WBD, and aim to identify evidence for specific factors underlying variation in dosing.

## Methods

Data was obtained from five PPS surveys in the Global Antimicrobial Resistance, Prescribing and Efficacy in Neonates and Children (GARPEC) study. Detailed methods have been described elsewhere [7]. Briefly, one 1-day pilot PPS was carried out over 2 months in 2015 and four full-scale 1-day PPSs were conducted between February 2016 and February 2017: February-March 2016 (1st PPS), May-June 2016 (2nd PPS), September-October 2016 (3rd PPS), and December 2016-February 2017 (4th PPS). The details of antibiotic prescription (drug, dose, route of administration), indication (targeted or empiric) and demographic data (e.g. sex, body weight), clinical indications and comorbidities (except in 65 of 162 children from the Pilot PPS) were collected. Patients within each PPS survey were anonymised with a unique identifier allowing identification of children receiving multiple antibiotics in the same survey. Age was recorded in months until 60 months and in years, thereafter; gestational age at birth was not captured. Data were collected between March 2015 and February 2017 and were fully anonymised before access for analysis. The PPS surveys were conducted as clinical audit and not routine health surveillance or research as the survey involved no interventions or experimentation; St George's University of London Research Ethics Committee provided

confirmation that ethical approval was not required for the initial survey (ARPEC) on which the PPS surveys were based. Consequently, formal ethical approval and/or written informed consent was not required in many of the 65 hospitals and no central ethical approval was obtained or Clinical Audit Facilitator was consulted. It was the responsibility of each of the participating hospitals received local ethics approval if required. The authors were not involved in local data collection and so did not access identifying patient information at any time. The data collection was carried out by local participating sites and they voluntarily contributed their data to GARPEC network.

## Data preparation

Data consisted of intravenous doses (recorded as parenteral in pilot PPS) given for treatment and not prophylaxis or decolonisation to hospitalised children with recorded age 1 month– 12 years old (as per Standards for Research (StaR) in Child Health project [16]) and where weight and sex was also recorded. We focused on the mostly frequently prescribed 16 of 76 antibiotics, each with at least 75 doses recorded, comprising 83% of all IV doses. Observations where the absolute value of WHO weight-for-age z-scores [16] was greater than eight standard deviations from the mean from 21 individuals up to 10 years old were excluded as likely data processing errors. For children aged 11–12 years, weights for six children outside 2.5 and 97.5 percentiles were replaced by the respective percentile values as height data was not available, and is required, to calculate weight z-scores for these older children. A further 2.9% of doses were excluded as extreme outliers (details in S1 Appendix and S1 and S2 Tables) and five doses recorded as less than once per day (e.g. every 36h) were counted as once per day.

## Analysis of dosing strategy

Dosing was analysed using two different metrics–fixed daily dose (FDD), in mg per day, and weight-based dose (WBD), in mg per kg per day, where WBD is equal to FDD divided by the child's weight in kg. Doses were assigned to being either consistent with 'standard FDD' only, consistent with 'standard WBD' only, consistent with 'both standard FDD and WBD' or consistent with 'neither'.

'Standard FDDs' were defined as doses given to at least 5% of children receiving each antibiotic, with the cut-off of 5% chosen following inspection of the frequency plot of doses. Standard FDDs were then compared to standard vial sizes from EMC [17] and BNFc [18].

'Standard WBDs' was less straightforward to define as the actual dose given to children may have been rounded to a dose in mg for more convenient administration. We could not assign standard WBDs statistically for each antibiotic using mixture models, as the models did not generally converge. Instead, WBD doses, in mg/kg/day, were rounded to the nearest whole number, apart from gentamicin, which was rounded to one decimal place. The top 10 WBDs for each antibiotic were then identified, including WBDs within 1% of the top WBDs as being associated with that WBD; this was done sequentially so that each dose was assigned to one standard WBD only. The top dose for metronidazole was taken to be 22.5mg/kg/day as this is commonly recommended [18–21]. As with FDD, frequency plots were used to determine the cut-off value for WBD, which was also 5%. WBD were compared to dosing recommendations for common indications from six sources—the USA 'Red Book' [19], the European 'Blue Book [20], the British National Formulary for children (BNFc) [18], the Indian National Centre for Disease Control [21], the WHO Pocket Book of Hospital Care in Children [22], and the summary of product characteristics (SPC) [17]. Guideline values for IV treatment were extracted from Mathur et al 2020 [23], except cefepime and teicoplanin, which were taken from the source documents, and SPCs which were obtained from the electronic medicines

compendium (EMC) website. To determine if some of the highest WBD doses were due to premature babies over one month of age we also plotted WBD for children over four months old (81% of doses), when all babies are expected to have a gestational age of at least 40 weeks.

## Analysis of factors underlying variation in WBD

Factors underling variation in WBD (mg/kg/day) were analysed truncating WBD for each antibiotic at its 2.5% and 97.5% percentiles to avoid undue influence from outliers, and using a Box-Cox transformation to achieve approximate normality (lambda = 0.109), which was then mean-centred for each antibiotic to facilitate comparison. No individual patient clinical data was available from these single day PPS. Within-individual explanatory fixed-effect variables included the top nine clinical indications for treatment above category 'other' (88% of doses); empiric (awaiting culture or no culture taken) or targeted (pathogen or resistance profile known) therapy; presence of comorbidities (none, renal, liver and other); and markers of likely serious infection such as ventilation status (no/non-invasive ventilation grouped vs invasive ventilation), taking other antibiotics (binary) and taking other drugs apart from antibiotics (binary). Grouping of predictor variables was determined based on prevalence of factors and consultation with a clinical (NR). Analyses were restricted to complete cases for explanatory variables (see Table 1 for numbers included in model). The correlation between weight and age was high (0.88) and so only age was included as it was not a component of the response variable. Age was centred on the median value and fitted including linear and quadratic terms and presented in units of 5 years for clarity.

We fitted the same linear mixed effects model for each antibiotic separately with independent between-individual random effects of country and hospital using the lme4 package [24]

**Table 1. Patient characteristics split by antibiotic.**

| Antibiotic | N | N in statistical model | Median age months (IQR) | Mean weight kg (SD) | Most common diagnosis | % infant, toddler, early childhood, middle childhood | % WHO Region Africa, Americas, Europe, South-East Asia, Western Pacific |
|---|---|---|---|---|---|---|---|
| Amikacin | 254 | 198 | 24 (7, 60) | 13 (10) | FN/ Fever (20%) | 36%, 10%, 30%, 24% | 5%, 37%, 38%, 19%, 2% |
| Ampicillin | 158 | 122 | 16 (3, 48) | 13 (11) | Bac LRTI (46%) | 48%, 12%, 20%, 20% | 20%, 44%, 20%, 4%, 13% |
| Cefepime | 196 | 175 | 24 (8, 60) | 13 (8) | FN/ Fever (26%) | 34%, 13%, 30%, 23% | 1%, 89%, 5%, 0%, 6% |
| Cefotaxime | 170 | 133 | 13 (3, 48) | 11 (9) | Sepsis (29%) | 49%, 9%, 26%, 15% | 2%, 5%, 54%, 24%, 15% |
| Ceftazidime | 103 | 86 | 36 (15, 72) | 16 (9) | Bac LRTI (40%) | 18%, 14%, 36%, 32% | 1%, 22%, 57%, 15%, 5% |
| Ceftriaxone | 472 | 352 | 24 (8, 60) | 14 (10) | Bac LRTI (31%) | 35%, 15%, 28%, 22% | 7%, 26%, 36%, 24%, 6% |
| Cefuroxime | 92 | 60 | 24 (11, 60) | 16 (12) | Bac LRTI (32%) | 30%, 15%, 30%, 24% | 1%, 21%, 77%, 1%, 0% |
| Ciprofloxacin | 74 | 61 | 36 (13, 96) | 16 (11) | Sepsis (30%) | 26%, 7%, 31%, 36% | 0%, 32%, 50%, 15%, 3% |
| Clindamycin | 109 | 78 | 36 (19, 84) | 20 (12) | SSTI (32%) | 18%, 11%, 37%, 34% | 1%, 51%, 39%, 3%, 6% |
| Co-amoxiclav | 263 | 198 | 24 (9, 60) | 14 (10) | Bac LRTI (53%) | 35%, 14%, 31%, 21% | 3%, 28%, 43%, 23%, 3% |
| Gentamicin | 215 | 180 | 20 (4, 72) | 15 (13) | Bac LRTI (21%) | 45%, 8%, 19%, 28% | 17%, 13%, 54%, 3%, 13% |
| Meropenem | 397 | 336 | 18 (6, 48) | 12 (9) | Sepsis (32%) | 44%, 11%, 25%, 20% | 7%, 27%, 44%, 21%, 2% |
| Metronidazole | 132 | 93 | 60 (24, 96) | 19 (12) | Surgical Dis (34%) | 20%, 5%, 36%, 40% | 5%, 27%, 48%, 16%, 5% |
| Pip-taz | 287 | 233 | 36 (14, 84) | 17 (12) | FN/ Fever (37%) | 23%, 13%, 34%, 30% | 3%, 15%, 68%, 10%, 5% |
| Teicoplanin | 83 | 59 | 48 (12, 90) | 18 (13) | FN/ Fever (41%) | 25%, 7%, 30%, 37% | 0%, 14%, 82%, 1%, 2% |
| Vancomycin | 362 | 294 | 16 (5, 48) | 12 (9) | Sepsis (27%) | 42%, 16%, 24%, 18% | 4%, 47%, 33%, 8%, 8% |

Bac LRTI: Bacterial lower respiratory tract infection, CNS: central nervous system infection; CRBSI: catheter related blood stream infection; FN: febrile neutropenia; GUTI: gastro-intestinal tract infections; SSTI: skin and soft tissue infections; Dis: disease; UTI: urinary tract infection; Infant: 1–12 months; Toddler 13–23 months; Early childhood: 2–5 years; Middle childhood: 6–11 years.

in R [25]. Box-Cox transformations used MASS package [26], significance testing of individual factor levels used Satterthwaite approximations from the lmerTest package [27], and 95% confidence intervals were obtained using the model-outputted t-statistic and degrees of freedom. Data used for analyses is available in S1 Data.

## Results

Data consisted of 3,367 doses from 2,463 children in 65 hospitals in 23 countries. Almost half (43%) of doses were from the WHO European Region, 31% the Americas, 14% South-East Asia, 6% Western Pacific Region and 5% African Region; none were from Eastern Mediterranean Region (Table 1). The median age of children was 24 months (IQR: 7–60 months, Range: 1 month–11 years) and 35% of doses were given to children under one year of age. The median weight of children was 11kg (IQR: 6.5–19.0 kg, Range: 0.74–64.8kg). Just over half of doses (54%) were given to boys.

### Frequency

Drugs varied considerably in the frequency of administration per day (Fig 1A and S3 Table). Amikacin, gentamicin and teicoplanin were predominantly (but not exclusively) given once per day whereas ampicillin and vancomycin were predominantly given 4 times per day. Ten of the 16 antibiotics had a clear main frequency given in at least 80% of doses. Three antibiotics (ceftriaxone, clindamycin, and piperacillin-tazobactam) had two common frequencies each given in approximately half of doses, and the remaining three antibiotics (cefotaxime, ciprofloxacin and vancomycin) had a predominant frequency given in between 60% and 65% of doses. Restricting data to children over four months (S1 Fig) removed some of the frequencies given to small numbers of children but they broad patterns remained the same.

### Dosing strategy

WBD clustered around a small number of peaks, and all antibiotics had between one and three standard WBDs used in at least 5% of doses (Fig 2 and S4 Table). Cefepime had the highest proportion of doses consistent with dosing by standard WBD, although this still only occurred in under half of doses (48%) with all of these doses being within 1% of 150mg/kg/day. Gentamicin, piperacillin-tazobactam and teicoplanin all had under 25% of doses consistent with dosing by standard WBD. Overall, 84% of doses were within the minimum and maximum values from the different guidelines and SPC. The antibiotic with the lowest percentage of doses within guideline ranges was vancomycin (46%), while the highest was ampicillin (99%). Restricting data to children over four months removed some of the extremely high WBD in amikacin, cefepime and teicoplanin but the broad patterns remained similar (S2 Fig).

Eleven of 16 antibiotics had between one and three standard FDDs that were administered in at least 5% of doses. Ceftriaxone and ciprofloxacin had the highest proportion of doses consistent with standard FDD at 30% and 28%, respectively (Fig 3 and S5 Table). However, teicoplanin had the highest percentage of doses (11%) given as the same single FDD (400mg). Five antibiotics (amikacin, cefepime, gentamicin, meropenem and piperacillin-tazobactam) had no standard FDDs that that were administered in at least 5% of doses. Of the 26 standard FDDs, 16 (62%) could be attributed to children just being given between one and three full vials of 100mg, 200mg, 250mg, 300mg, 500mg, 600mg, 1g, and 2g, depending on the vial sizes available for each drug. Standard FDD that we could not attribute to a small number of full vials were cefotaxime 1200mg, ceftazidime 3600mg, cefuroxime 1800mg and 4500mg, clindamycin 120mg, co-amoxiclav 600mg (perhaps 500mg amoxicillin and 100mg clavulanate), metronidazole 150mg and 600mg, teicoplanin 160mg and vancomycin 600mg.

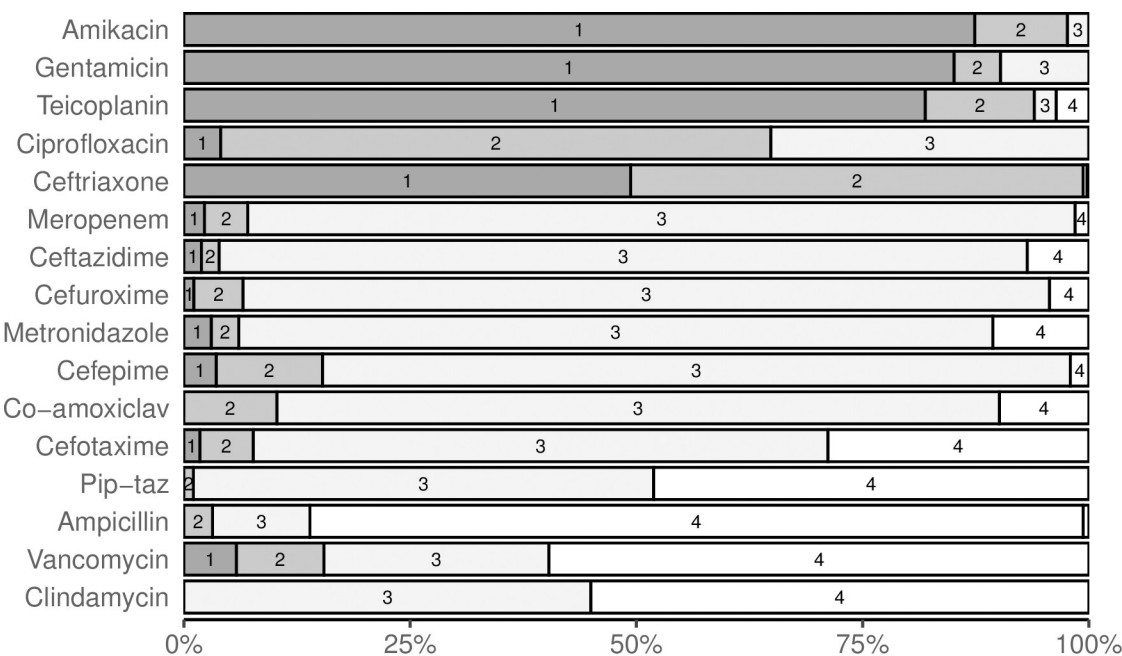

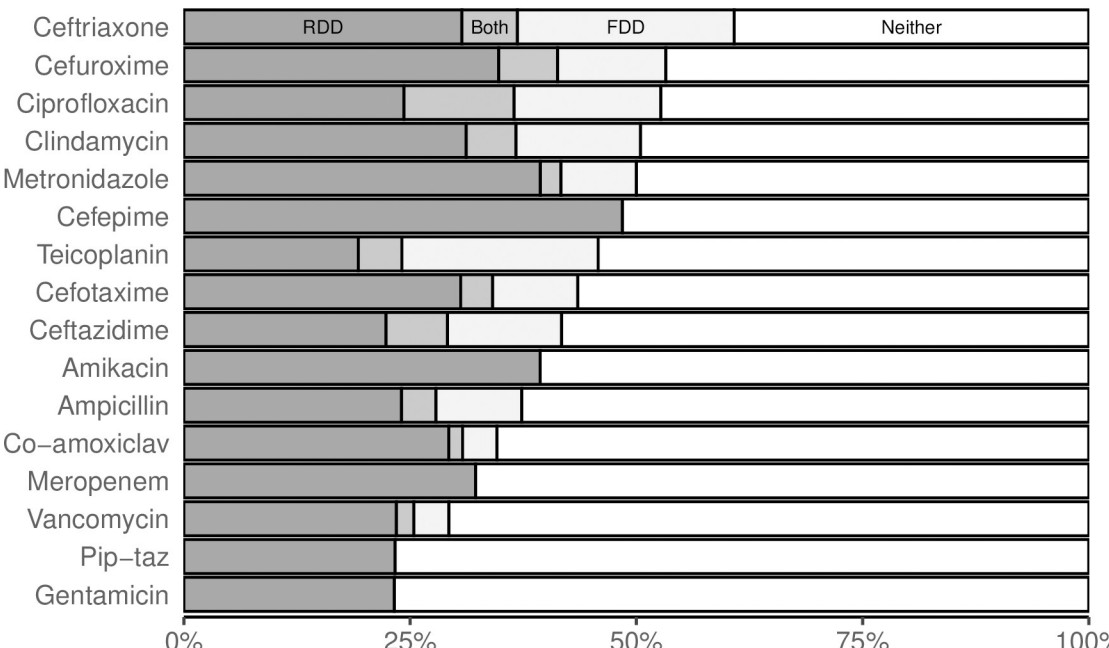

**Fig 1. Frequency and dosing strategy by antibiotic.** A. Frequency of antibiotic dosing within 24h. Antibiotics are ordered by most common frequency–once per day at the top and 4 times per day at the bottom. Note that frequency once per day includes 5 doses given less than once per day (e.g. every 36h). B. Dosing strategy for each antibiotic. Each dose was assigned to being consistent with dosing by fixed daily dose (FDD), weight-based dose (WBD), both or neither. Antibiotics are sorted by the proportion assigned to a dosing strategy.

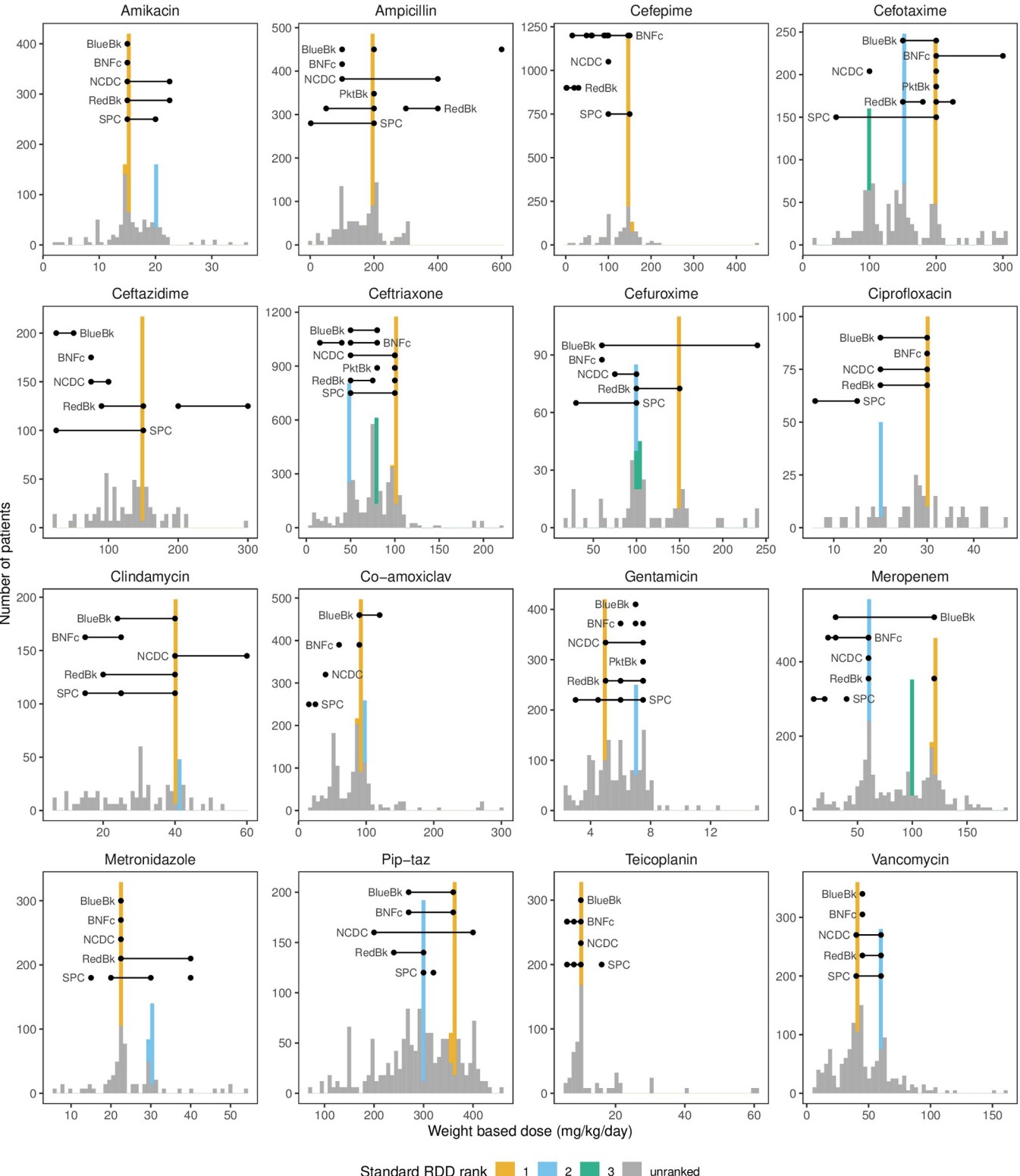

**Fig 2. Antibiotics showed between one and three peaks (coloured bars) consistent with dosing by standard WBD.** Overlaid are guideline recommendations for IV dosing in children, generally and for priority syndromes (severe infection, pneumonia, sepsis, acute otitis media, pharyngitis, urinary tract infection) from five different guidelines: Blue Book (BlueBk), British National Formulary for children (BNFc), Indian National Centre for Disease Control (NCDC), WHO pocket book (PktBk) and Red book (RedBk). Multiple entries for a single guideline indicate different frequencies of daily dosing, indications (e.g.

meningitis) or age/weight group of children. Lines denote recommended range, and dots denote recommended dose. Seven recommendations of fixed daily doses (mainly for larger children) have been converted to relative daily doses. Bars coloured both gray and ranked are due to rounding used to produce graph.

The proportion of doses assigned to a dosing strategy (standard WBD and/or standard FDD) varied considerably, from 23% for piperacillin-tazobactam to 61% for ceftriaxone FDD (Fig 1B and S6 Table). Ceftriaxone, cefuroxime, ciprofloxacin, clindamycin and metronidazole had over half of doses consistent with dosing by either WBD or FDD, whereas gentamicin, meropenem, piperacillin-tazobactam and vancomycin all had less than of one third of doses consistent with dosing by WBD. All drugs had a higher proportion of doses consistent with WBD than FDD, apart from teicoplanin, which had approximately 25% of doses consistent

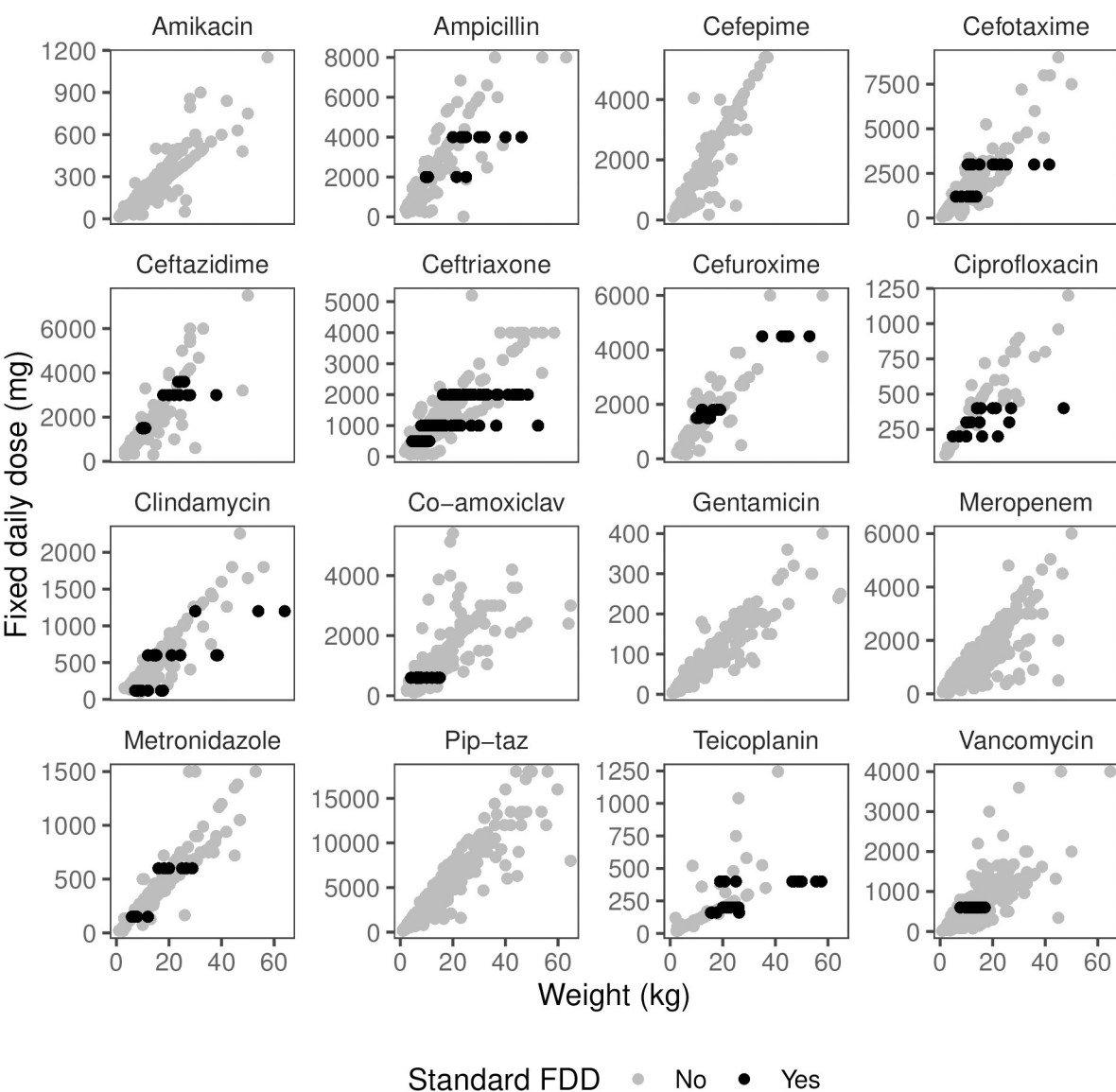

**Fig 3. Antibiotics varied in their use of standard fixed doses (FDD).** Black dots show doses in mg shared by at least 5% of children. Grey dots show all other doses–the doses following diagonal lines are constant mg/kg/day.

with each strategy. All drugs had less than 15% of doses consistent with dosing by both WBD and FDD.

Drugs varied in their pattern of dosing strategy of (standard WBD and/or standard FDD) by both weight and age, although no clear patterns emerged (S3 and S4 Figs). For example, cefepime dosing was broadly consistent across all weight and age bands; teicoplanin and ceftriaxone showed increasing tendency towards dosing by FDD with increasing weight and age; and metronidazole showed increasing tendency of dosing by WBD with increasing weight and age.

### Factors underlying variation in WBD

Models consisted of 2,658 doses across 16 antibiotics. The sample size in each model ranged from 59 for Teicoplanin to 352 for Ceftriaxone (median 154; Table 1).

There were no strong consistent patterns in associations between factors and dosing across antibiotics. Compared to bacterial lower respiratory tract infections, dosing was significantly lower for patients with skin and soft tissue infections for four antibitoics (ceftazidime, co-amoxiclav, meropenem and piperacillin-tazobactam), and significantly higher for patients with central nervous system infections for three antibiotics (cefotaxime, meropenem, vancomycin; Fig 4 and S7 Table), although amikacin was associated with significantly lower dosing for the same diagnosis. Lower average WBD for children with renal comorbidities compared to no morbidities was observed for cefepime, ceftazidime, meropenem and vancomycin. Three antibiotics, cefotaxime, gentamicin and vancomycin, were associated with lower WBD in hospital acquired infections compared to community acquired infections; vancomycin was also associated with higher WBD when given targeted than empirically.

The statistical models showed strikingly different patterns in the variation remaining after adjusting for the effects of the factors considered (S5 Fig and S8 Table). In particular, clindamycin showed substantial proportions of variation due to country (42%) and hospital (34%). In contrast, cefepime and teicoplanin showed no evidence of consistent variation between countries or between hospitals.

## Discussion

We characterised variation in antibiotic dosing for IV treatment in children using PPS data from over three thousand prescriptions in 65 hospitals across 23 countries. Dosing predominantly clustered around a small number of strategies, although variation remained. There were no clear association between the patterns of dosing and comorbidities, underlying disease, or clinical infection severity, apart from some antibiotics having higher doses in central nervous system infections, and lower doses in skin and soft tissue infections compared to lower respiratory tract infections (the most common indication).

The majority of antibiotics (10/16) had a common frequency of administration given in at least 80% of doses that were broadly consistent with PKPD principles. For instance, the two aminoglycosides, amikacin and gentamicin, were predominantly given once per day, consistent with concentration-dependent killing, whereas cephalosporins and penicillins were predominantly given three or four times daily consistent with time-dependent killing [10].

Dosing strategy was attributable to weight-based dosing (WBD) more often than fixed daily dosing (FDD, e.g. by age band) for all antibiotics apart from teicoplanin, which had approximately equal proportions of dosing attributable to each strategy. Dosing by weight, rather than fixed daily doses (perhaps split by age), may reflect the predominance of weight-based dosing in most international guidelines and enables fine-scale alterations to dosing, for example when renal comorbidities are present. FDD may be more straightforward to administer by vial but

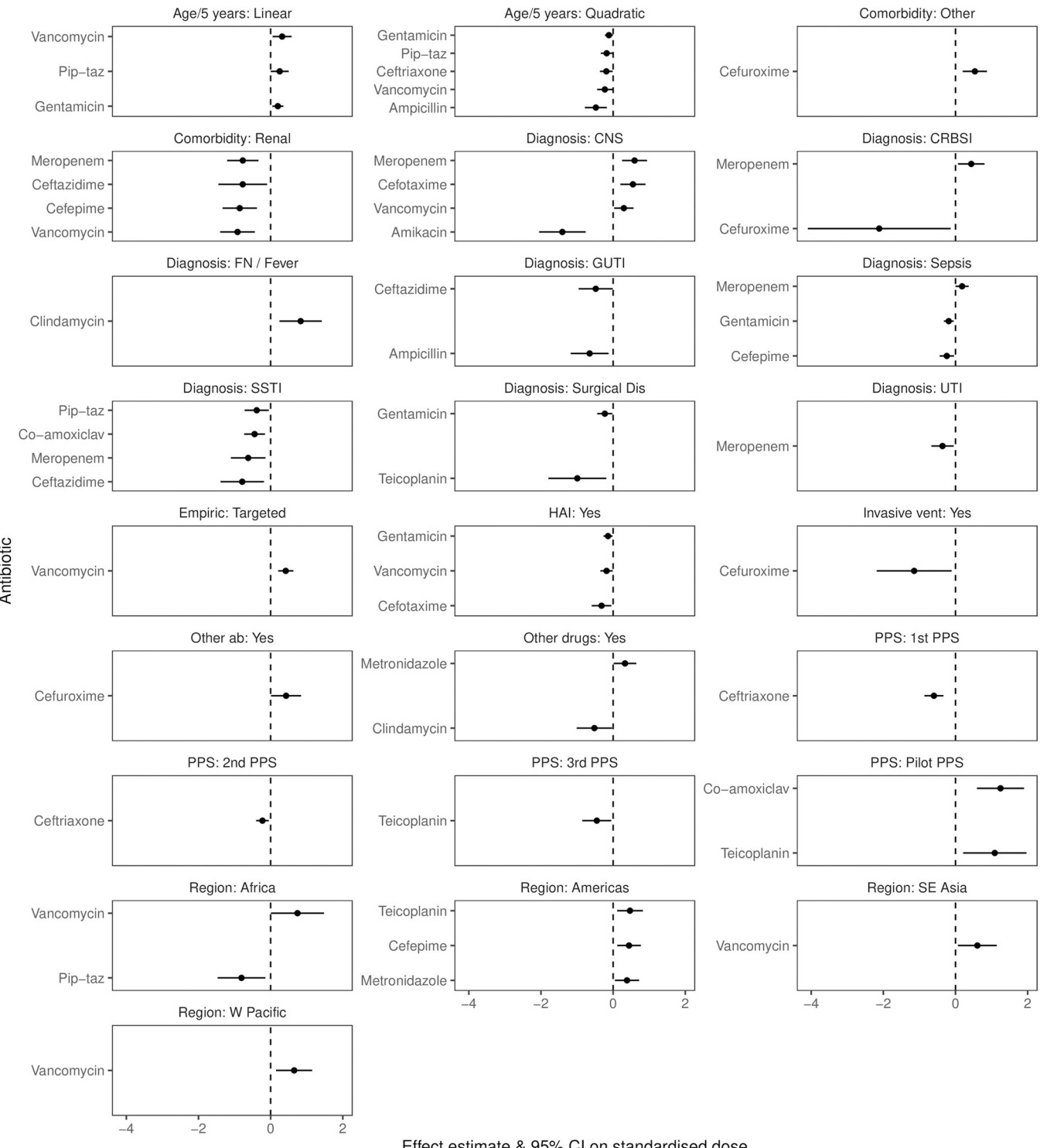

**Fig 4. Effect estimates and 95% confidence intervals of significant effects (p<0.05) from models of standardised WBD for the most common diagnoses.**
Models were run separately for each antibiotic and contained the same fixed and random effects for each model. Reference levels for each factor were chosen as the level with the highest proportion of doses (region: Europe, PPS: most recent survey (4th), diagnosis: bacterial LRTI, empiric: empiric, sex: male). CNS: central nervous system infection; CRBSI: catheter related blood stream infection; FN: febrile neutropenia; GUTI: gastro-intestinal tract infections; SSTI: skin and soft tissue infections; Dis: disease; UTI: urinary tract infection; HAI: hospital acquired infection; Vent: ventilation; ab: antibiotic; PPS: point prevalence survey; SE: south-east; W: west.

risks over or under-dosing children, particularly those of extreme weight for age, potential negative consequences in terms of efficacy, toxicity and antimicrobial resistance. Dosing in WBD clustered around one to three standard doses for each antibiotic. These were broadly consistent with one or more of the six current guidelines we examined, but not all, suggesting that guidelines are taken into consideration when making dosing decisions, although comparisons are complicated as guidelines themselves show different levels of consistency for different antibiotics. However, between approximately 40% and 80% of doses for each antibiotic were not attributable to standard dosing by FDD or WBD. This may be due to use of local hospital guidelines, perhaps reflected in the substantial residual country-to-country and hospital-to-hospital variation estimates for some antibiotics after accounting for child-level characteristics, or dosing decisions taken by individual prescribers.

Dosing was attributable to FDD in up to 30% of doses for each antibiotic. In theory, this could be partly due to older, heavier children receiving adult doses, but we found no strong evidence for this across antibiotics. Approximately 60% of FDDs could be attributed to simple multiples of vial sizes, suggesting that dosing strategy could simply be reflecting vial sizes available at the point of prescription.

Variation in dosing might be expected due to varying PKPD properties in different clinical scenarios, such as diagnosis, TDM (mainly amikacin, vancomycin and gentamicin) or variation in antibiotic susceptibility [10, 28]. Although this seemed to be the case for some drugs, such as dosing relative to weight was higher for vancomycin in targeted therapy than empiric therapy, the high proportion of vancomycin doses outside of guidelines, and higher dosing for central nervous system infections than lower respiratory tract infections, we did not find evidence of strong patterns across all antibiotics. The finding of lower WBD on average in hospital acquired infections compared to community acquired infections for cefotaxime, gentamicin and vancomycin was curious and not easily explainable, emphasising the need for closer attention to be paid to dosing of antibiotics in children.

This study has several limitations. Firstly, we did not have any data on loading doses, bolus, short infusion, prolonged infusion, different doses on start or stop days, or TDM (expected to affect only amikacin, vancomycin and gentamicin), which may be driving some of the differences observed. Secondly, gestational age was not captured and some extremely pre-term babies may be receiving neonatal doses despite being at least one month old, although patterns in data restricted to babies over four months old (at least term) did not differ substantially from the full dataset. Thirdly, the statistical analyses used different numbers of observations (between 59 and 352; median of 154) for each antibiotic, and a relatively large number of statistical tests. Results should therefore be interpreted in this context and conclusions drawn on broad patterns only. Finally, although the survey included 23 countries, not all WHO regions were equally represented. Specifically, there were no data from Eastern Mediterranean Region and almost half of doses came from European region. Although both Africa and Western Pacific regions each accounted for approximately 5% of doses, both regions were relatively well represented in ampicillin and gentamicin, with approximately 17% of doses from Africa region and 13% of doses from Western Pacific region. Additionally, data came from four hospitals in Africa region and seven in Western Pacific region (compared to 33 in Europe region), lending generalisability to our findings.

Antibiotic stewardship interventions around optimal dosing in children is hampered by inconsistency and lack of an evidence base. This affects not only prescribing practice, but also research, as there are no clearly accepted standard international guidelines for comparators in trials. We can identify no clear rationale for the historical variation in dosing guidance internationally. There is a clear opportunity for the clinical, scientific and public health communities to consolidate behind a consistent set of dosing guidelines for practice and future research

priorities. We suggest the relevant organisations that produce widely used antibiotic guidelines for children have an opportunity to harmonise international guidance for antibiotic dosing in children based on best available evidence [1] and knowledge of current practice presented here.

## Supporting information

**S1 Fig. Daily frequency of antibiotic dosing in children over four months.**
(PNG)

**S2 Fig. Weight based dosing (mg/kg/day) in children over 4 months old.** Highlighted common doses are taken from the original analyses on all children.
(PNG)

**S3 Fig. Proportion of doses consistent with daily dosing and relative daily dosing by weight quintiles for each antibiotic.** Doses consistent with both strategies are included in both categories.
(PNG)

**S4 Fig. Dosing strategy by age group for each antibiotic.** Infant: 1–12 months; Toddler 13–23 months; Early childhood: 2–5 years; Middle childhood: 6–11 years.
(PNG)

**S5 Fig. Variance components from models of RDD.** Models were run separately for each antibiotic and contained the same fixed and random effects for each model. Random effects were country and hospital within country.
(PNG)

**S1 Appendix. Additional details on clearning of dosing data.**
(DOCX)

**S1 Data.**
(CSV)

**S1 Table. Thresholds for identifying extreme outliers and number of excluded observations for each antibiotic.**
(DOCX)

**S2 Table. Summary of doses included in dosing strategy analyses by antibiotic.**
(DOCX)

**S3 Table. Antibiotic frequency in 24h period by antibiotic.** Note that frequency once per day includes 5 doses given less than once per day (e.g. every 36h).
(DOCX)

**S4 Table. Top RDD doses for each antibiotic (percentage of doses).** Doses measured in mg/kg/day, including include a buffer of ±1% to allow for rounding before administration, and given in at least 5% of cases.
(DOCX)

**S5 Table. Top FDD doses for each antibiotic (percentage of doses).** Doses measured in mg/day and given in at least 5% of cases.
(DOCX)

**S6 Table. Dosing strategy split by antibiotic.**
(DOCX)

**S7 Table. Fixed effects estimates from models of RDD.** Models were run separately for each antibiotic and contained the same fixed and random effects for each model. Random effects were country and hospital within country.
(DOCX)

**S8 Table. Variance components estimates from models of RDD.** Models were run separately for each antibiotic and contained the same fixed and random effects for each model. Random effects were country and hospital within country.
(DOCX)

## Acknowledgments

We thank all of the members of the GARPEC network for their participation and GARDP and PENTA for their support of the project. We also thank Ann Versporten for thoughtful comments.

## Author Contributions

**Conceptualization:** Julia A. Bielicki, Yingfen Hsia, Mike Sharland.

**Data curation:** Michelle N. Clements, Neal Russell, Julia A. Bielicki, Sally Ellis, Yingfen Hsia, Mike Sharland.

**Formal analysis:** Michelle N. Clements.

**Supervision:** Neal Russell, Julia A. Bielicki, Sally Ellis, Silke Gastine, Yingfen Hsia, Joseph F. Standing, A. Sarah Walker, Mike Sharland.

**Visualization:** Michelle N. Clements.

**Writing – original draft:** Michelle N. Clements.

**Writing – review & editing:** Michelle N. Clements, Neal Russell, Julia A. Bielicki, Sally Ellis, Silke Gastine, Yingfen Hsia, Joseph F. Standing, A. Sarah Walker, Mike Sharland.

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
