## [Decision Letter · Decision Letter 0]

9 Apr 2021

PONE-D-20-30233

Global antibiotic dosing strategies in hospitalised children: characterising variation and implications for harmonisation of international guidelines

PLOS ONE

Dear Dr. Clements,

Thank you for submitting your manuscript to PLOS ONE. After careful consideration, we feel that it has merit but does not fully meet PLOS ONE’s publication criteria as it currently stands. Therefore, we invite you to submit a revised version of the manuscript that addresses the points raised during the review process.

We look forward to receiving your revised manuscript.

Kind regards,

Monika Pogorzelska-Maziarz

Academic Editor

PLOS ONE

Journal Requirements:

2. In the ethics statement in the manuscript and in the online submission form, please provide additional information about the patient records used in your retrospective study, including: a) whether all data were fully anonymized before you accessed them; b) the date range (month and year) during which patients' medical records were accessed. If patients provided informed written consent to have data from their medical records used in research, please include this information.

3. Thank you for including your ethics statement: "The PPS surveys were considered a clinical audit and each participating hospital received local ethics approval if required."   

 a.Please amend your current ethics statement to include the full name of the ethics committee/institutional review board(s) that approved your specific study.

 b.Once you have amended this/these statement(s) in the Methods section of the manuscript, please add the same text to the “Ethics Statement” field of the submission form (via “Edit Submission”).

4. We note that the PPS surveys analysed in your study are described as a clinical audit. In your Methods section, please state whether a Clinical Audit Facilitator was consulted.

5. To comply with PLOS ONE submission guidelines, in your Methods section, please provide additional information regarding your statistical analyses. For more information on PLOS ONE's expectations for statistical reporting, please see https://journals.plos.org/plosone/s/submission-guidelines.#loc-statistical-reporting.

6. Thank you for stating the following in your Competing Interests section: 

"No"

potential competing interests for the purposes of transparency. PLOS defines a competing interest as anything that interferes with, or could reasonably be perceived as interfering with, the full and objective presentation, peer review, editorial decision-making, or publication of research or non-research articles submitted to one of the journals. Competing interests can be financial or non-financial, professional, or personal. Competing interests can arise in relationship to an organization or another person. Please follow this link to our website for more details on competing interests: http://journals.plos.org/plosone/s/competing-interests

7.  We note that you have indicated that data from this study are available upon request. PLOS only allows data to be available upon request if there are legal or ethical restrictions on sharing data publicly. For information on unacceptable data access restrictions, please see http://journals.plos.org/plosone/s/data-availability#loc-unacceptable-data-access-restrictions.

Reviewers' comments:

Reviewer's Responses to Questions

**Comments to the Author**

1. Is the manuscript technically sound, and do the data support the conclusions?

Reviewer #1: Yes

Reviewer #2: Partly

2. Has the statistical analysis been performed appropriately and rigorously? 

Reviewer #1: I Don't Know

Reviewer #2: I Don't Know

3. Have the authors made all data underlying the findings in their manuscript fully available?

Reviewer #1: Yes

Reviewer #2: Yes

4. Is the manuscript presented in an intelligible fashion and written in standard English?

Reviewer #1: Yes

Reviewer #2: Yes

5. Review Comments to the Author

Reviewer #1: I read with interest the article " Global antibiotic dosing strategies....." by Clements et al.

It is an engaging article that tries to identify factors behind variation in antibiotic dosing using data derived from 5 global point prevalence surveys.

They are unable to identify strong consistent pattern behind historical variation in weight based dosing except higher doses are used in CNS infection and lower in cutaneous and soft tissue infection compared to LRTI.

Presentation of result is clear and readability of discussion is good. Limitations are clearly depicted.

Overall this is an easy to read paper, free of grammatical errors and has a meaningful impact.

Reviewer #2: In this manuscript the authors use data from five global point prevalence surveys to examine the variation in antibiotic dosing approaches in children 1 month to 12 years. This is an interesting paper and the authors effectively illustrate the significant variation in guidelines and need for harmonisation.

Major comments: I think this study would better be done by excluding those infants <1 year due to the major limitation that the authors acknowledge of not having data on gestational age. It is clear from some of the dosing intervals that premature babies have been included with some drugs dosed 36-hourly dosing and 8% of cefotaxime doses being once or twice daily. I expect that you will see less variation if you were to exclude those infants <1 year of age.

Also, it appears in the supplementary material that the effect of specific comorbidities i.e. liver or renal disease is not analysed separately. I think this would be important to do as doses are often adjusted with these comorbidities as they affect drug clearance.

Minor comments:

Line 46 – ‘despite difference in PKPD’ are there any studies showing that the PD is different in children compared to adults? Suggest change to PK only

Line 89 – ‘Children aged 11-12 years, weights for six 88 children outside 2.5 and 97.5 percentiles were truncated at the respective values as height data was not available, and is required, to calculate weight z-scores for these older children’ – I’m unclear what the authors mean by ‘truncated at the respective values’

Line 110 – Metronidazole top dose is 22.5mg/kg/day as this is commonly recommended. This is an unusually high IV metronidazole dose. Can the authors provide a reference for this?

Line 117 – Please define EMC

Line 126 – Presence of comorbidities – need to look specifically at liver or renal disease which is likely to reflect drug clearance. It appears that all comorbidities have been analysed together

Table 1 – The column % infant, toddler, early childhood, middle childhood does not add additional useful information above median age and IQR. Suggest remove.

Line 218 – The methods state that ‘correlation between weight and age was high (0.88) 131 and so only age was included as it was not a component of the response variable.’ However, later it states that ‘Drugs varied in their pattern of dosing strategy by weight and age’. Can you please clarify whether both weight and age analysis was done?

Line 232 - Three antibiotics, cefotaxime, gentamicin 233 and vancomycin, were associated with lower WBD in hospital acquired infections compared 234 to community acquired infections; - this is an unusual finding, can the authors comment on why this would be?

Limitations: It would be important to highlight that the data are primarily from a cohort of young children (75% are 5 years or younger)

Discussion - the paper places a lot of emphasis on FDD versus RDD but there is no discussion as to the pros/cons of these strategies. What do the authors feel is the best approach?

6. PLOS authors have the option to publish the peer review history of their article (what does this mean?). If published, this will include your full peer review and any attached files.

Reviewer #1: **Yes: **Amit P. Ladani

Reviewer #2: No

---

## [Author Response · Author response to Decision Letter 0]

11 May 2021

Journal requirements 

Updated

2. In the ethics statement in the manuscript and in the online submission form, please provide additional information about the patient records used in your retrospective study, including: a) whether all data were fully anonymized before you accessed them; b) the date range (month and year) during which patients' medical records were accessed. If patients provided informed written consent to have data from their medical records used in research, please include this information. 

And

 3. Thank you for including your ethics statement: "The PPS surveys were considered a clinical audit and each participating hospital received local ethics approval if required." 

 a.Please amend your current ethics statement to include the full name of the ethics committee/institutional review board(s) that approved your specific study.

 b.Once you have amended this/these statement(s) in the Methods section of the manuscript, please add the same text to the “Ethics Statement” field of the submission form (via “Edit Submission”).

And

 4. We note that the PPS surveys analysed in your study are described as a clinical audit. In your Methods section, please state whether a Clinical Audit Facilitator was consulted.

The ethics section of the manuscript (lines 77-83) now reads: 

“Data were collected between March 2015 and February 2017 and were fully anonymised before access for analysis. The PPS surveys were conducted as clinical audit and not routine health surveillance or research as the survey involved no interventions or experimentation; St George’s University of London Research Ethics Committee provided confirmation that ethical approval was not required for the initial survey (ARPEC) on which the PPS surveys were based. Consequently, formal ethical approval and/or written informed consent was not required in many of the 65 hospitals and no central ethical approval was obtained or Clinical Audit Facilitator was consulted. It was the responsibility of each of the participating hospitals received local ethics approval if required. The authors were not involved in local data collection and so did not access identifying patient information at any time. The data collection was carried out by local participating sites and they voluntarily contributed their data to GARPEC network.”

 5. To comply with PLOS ONE submission guidelines, in your Methods section, please provide additional information regarding your statistical analyses. For more information on PLOS ONE's expectations for statistical reporting, please see https://journals.plos.org/plosone/s/submission-guidelines.#loc-statistical-reporting.

Additional information has been added to the methods section

6. Please complete your Competing Interests on the online submission form to state any Competing Interests. If you have no competing interests, please state "The authors have declared that no competing interests exist.", as detailed online in our guide for authors at http://journals.plos.org/plosone/s/submit-now. This information should be included in your cover letter; we will change the online submission form on your behalf.potential competing interests for the purposes of transparency. PLOS defines a competing interest as anything that interferes with, or could reasonably be perceived as interfering with, the full and objective presentation, peer review, editorial decision-making, or publication of research or non-research articles submitted to one of the journals. Competing interests can be financial or non-financial, professional, or personal. Competing interests can arise in relationship to an organization or another person. Please follow this link to our website for more details on competing interests: http://journals.plos.org/plosone/s/competing-interests

The authors have declared that no competing interests exist in the relevant sections

7. We note that you have indicated that data from this study are available upon request. PLOS only allows data to be available upon request if there are legal or ethical restrictions on sharing data publicly. For information on unacceptable data access restrictions, please see http://journals.plos.org/plosone/s/data-availability#loc-unacceptable-data-access-restrictions. In your revised cover letter, please address the following prompts:

 b) If there are no restrictions, please upload the minimal anonymized data set necessary to replicate your study findings as either Supporting Information files or to a stable, public repository and provide us with the relevant URLs, DOIs, or accession numbers. Please see http://www.bmj.com/content/340/bmj.c181.long for guidelines on how to de-identify and prepare clinical data for publication. For a list of acceptable repositories, please see http://journals.plos.org/plosone/s/data-availability#loc-recommended-repositories. We will update your Data Availability statement on your behalf to reflect the information you provide.

 Anonymised data has been added to the supplementary information. 

 Done

Reviewers’ comments

Major comments: 

1. I think this study would better be done by excluding those infants <1 year due to the major limitation that the authors acknowledge of not having data on gestational age. It is clear from some of the dosing intervals that premature babies have been included with some drugs dosed 36-hourly dosing and 8% of cefotaxime doses being once or twice daily. I expect that you will see less variation if you were to exclude those infants <1 year of age.

Thank you for your comment. We agree that premature babies over one month of age might be skewing some of the frequency and weight based dosing analyses. However, we feel that removing all children under on years of age (32% of doses) is perhaps a too strict cut-off. We have included graphs for frequency and weight based dosing for children over four months of age in the supplementary information, removing 19% of doses. At five months of age even the most premature babies should have a gestational age of at least 40 weeks and so we feel this is an appropriate cut-off that preserves the maximum amount of data. We have made the following edits to the text:

Line 118: To determine if some of the highest WBD doses were due to premature babies over one month of age we also plotted WBD for children over four months old (81% of doses), when all babies are expected to have a gestational age of at least 40 weeks.

Lines 164:‘Restricting data to children over four months (Supplementary S1 Fig) removed some of the frequencies given to small numbers of children but they broad pattern remained the same.’

Line 183: ‘Restricting data to children over one year removed some of the extremely high WBD in amikacin, cefepime and teicoplanin but the broad patterns remained similar (Supplementary S2 Fig).

Line 312-314: ’Secondly, gestational age was not captured and some extremely pre-term babies may be receiving neonatal doses despite being at least one month old, although patterns in data restricted to babies over four months old (at least term) did not differ substantially from the full dataset.’

2. Also, it appears in the supplementary material that the effect of specific comorbidities i.e. liver or renal disease is not analysed separately. I think this would be important to do as doses are often adjusted with these comorbidities as they affect drug clearance.

Excellent point, thank you for pointing it out. We have updated the comorbidities factor in the analyses to include ‘renal’, ‘liver’, ‘other’ and ‘none’. We have updated the analyses including figure 4 and in the supplementary information. Alterations to the main text are:

Line 135 (fixed effects in models): ‘presence of comorbidities (none, renal, liver and other)’

Line 249-251: Lower average WBD for children with renal comorbidities compared to no morbidities was observed for cefepime, ceftazidime, meropenem and vancomycin.

Minor comments:

1. Line 46 – ‘despite difference in PKPD’ are there any studies showing that the PD is different in children compared to adults? Suggest change to PK only

Done

2. Line 89 – ‘Children aged 11-12 years, weights for six 88 children outside 2.5 and 97.5 percentiles were truncated at the respective values as height data was not available, and is required, to calculate weight z-scores for these older children’ – I’m unclear what the authors mean by ‘truncated at the respective values’

Have changed to “For children aged 11-12 years, weights for six children outside 2.5 and 97.5 percentiles were replaced by the respective percentile values as height data was not available, and is required, to calculate weight z-scores for these older children.”

3. Line 110 – Metronidazole top dose is 22.5mg/kg/day as this is commonly recommended. This is an unusually high IV metronidazole dose. Can the authors provide a reference for this?

Have added appropriate references

4. Line 117 – Please define EMC

Added ‘electronic medicines compendium’

5. Line 126 – Presence of comorbidities – need to look specifically at liver or renal disease which is likely to reflect drug clearance. It appears that all comorbidities have been analysed together

See major comment 2 above

6. Table 1 – The column % infant, toddler, early childhood, middle childhood does not add additional useful information above median age and IQR. Suggest remove.

We agree that frequencies and averages often do not need to appear together. However, we feel that both are valuable in this instance to improve interpretation for different types of reader, most statistically or clinically focused, and so we would be grateful if we could keep both in. 

Added 35% of doses were given to children under one year of age to line 145. 

7. Line 218 – The methods state that ‘correlation between weight and age was high (0.88) 131 and so only age was included as it was not a component of the response variable.’ However, later it states that ‘Drugs varied in their pattern of dosing strategy by weight and age’. Can you please clarify whether both weight and age analysis was done?

Apologies, we weren’t very clear here. The statistical models of mg/kg/day used only age as a predictor variable due to the high correlation between weight and age. However, when looking at the dosing strategy of fixed daily dosing (mg/day) and/or relative daily dosing (mg/kg/day) we looked for patterns across both age and weight groups, which is included in the supplementary information. We have updated the sentence to hopefully provide more clarity. 

Line 234 now reads: “Drugs varied in their pattern of dosing strategy of (standard WBD and/or standard FDD) by both weight and age, although no clear patterns emerged (Supplementary figs 1 & 2).”

8. Line 232 - Three antibiotics, cefotaxime, gentamicin 233 and vancomycin, were associated with lower WBD in hospital acquired infections compared 234 to community acquired infections; - this is an unusual finding, can the authors comment on why this would be?

Line 310-313 now reads: The finding of lower WBD on average in hospital acquired infections compared to community acquired infections for cefotaxime, gentamicin and vancomycin was curious and not easily explainable, emphasising the need for closer attention to be paid to dosing of antibiotics in children.

9. Limitations: It would be important to highlight that the data are primarily from a cohort of young children (75% are 5 years or younger)

Line 155-157 now reads: ‘The median age of children was 24 months (IQR: 7–60 months, Range: 1 month–11 years) and 35% of doses were given to children under one year of age.’

10. Discussion - the paper places a lot of emphasis on FDD versus RDD but there is no discussion as to the pros/cons of these strategies. What do the authors feel is the best approach?

Line 287-293 now reads: ‘Dosing by weight, rather than fixed daily doses (perhaps split by age), may reflect the predominance of weight-based dosing in most international guidelines and enables fine-scale alterations to dosing, for example when renal comorbidities are present. FDD may be more straightforward to administer by vial but risks over or under-dosing children, particularly those of extreme weight for age, potential negative consequences in terms of efficacy, toxicity and antimicrobial resistance. ‘

---

## [Editor Report · Decision Letter 1]

12 May 2021

Global antibiotic dosing strategies in hospitalised children: characterising variation and implications for harmonisation of international guidelines

PONE-D-20-30233R1

Dear Dr. Clements,

We’re pleased to inform you that your manuscript has been judged scientifically suitable for publication and will be formally accepted for publication once it meets all outstanding technical requirements.

Kind regards,

Monika Pogorzelska-Maziarz

Academic Editor

PLOS ONE

---

## [Editor Report · Acceptance letter]

19 May 2021

PONE-D-20-30233R1 

Global antibiotic dosing strategies in hospitalised children: characterising variation and implications for harmonisation of international guidelines 

Dear Dr. Clements:

I'm pleased to inform you that your manuscript has been deemed suitable for publication in PLOS ONE. Congratulations! Your manuscript is now with our production department. 

Kind regards, 

on behalf of

Dr. Monika Pogorzelska-Maziarz 

Academic Editor

PLOS ONE